# Dietary Plant Extracts Improve the Antioxidant Reserves in Weaned Piglets

**DOI:** 10.3390/antiox10050702

**Published:** 2021-04-29

**Authors:** Carlo Corino, Michel Prost, Barbara Pizzi, Raffaella Rossi

**Affiliations:** 1Department of Veterinary Medicine, Università degli Studi di Milano, Via Dell’Università 6, 26900 Lodi, Italy; raffaella.rossi@unimi.it; 2Lara Spiral SA, 21560 Couternon, France; michelprost.spiral@wanadoo.fr; 3Ausl di Parma, 43125 Parma, Italy; barbara.pizzi8@gmail.com

**Keywords:** plant extract, weaned piglets, antioxidant reserves

## Abstract

Reducing the use of antibiotics in livestock in order to contain antibiotic resistance and studying natural substance additives are key to sustainability. Among the various biological activities of plant extracts, antioxidant activity plays an important role. The present study assesses the total antioxidant activity and antioxidant reserves using the Kit Radicaux Libres test (KRL™ Kirial International, Couternon, France). One hundred and sixty piglets (Topics × Tempo) weaned at 28 days of age were divided into four dietary treatment groups that were fed a commercial diet (the control group, C); 500 mg/kg *Boswellia* extract (BOS); 200 and 50 mg/kg *Uncaria* and *Tanacetum* extracts (UT) respectively; and 225 mg/kg of an antioxidant plant extract mixture (AOX). The blood antioxidant activity of the piglets was measured using the KRL test and the reserves were analyzed on whole blood samples after hydrolysis with glucosidase, sulfatase and glucuronidase. No significant differences were observed in growth performance. The delta KRL values of the whole blood showed a significantly higher total antioxidant status of the piglets from the BOS and AOX groups than the UT and C groups (+30.7 BOS; +27.7 AOX vs. +17.81 UT +13.30 C; *p* = 0.002) between 18 and 28 days post-weaning. The delta KRL values of red blood cells (RBCs) showed a significantly higher total antioxidant status of the piglets from the AOX groups than the UT and BOS groups (+22.2 AOX; vs. +9.90 UT +9.4 BOS; *p* = 0.016) between the two sampling times. Reserves of UT and AOX were higher than C and BOS for all enzymes, glucosides, sulphates, and glucuronides. The biological KRL test proved to be an extremely sensitive tool to evaluate the piglets’ antioxidant status. Determining the antioxidant reserve also provides a better understanding of the real antioxidant status of pigs.

## 1. Introduction

Natural extracts are used in animal nutrition as sustainable additives to improve growth performance and health [1,2]. The European Union’s ban on the use of antibiotic growth promoters, together with the need to reduce the amount of antibiotics used in order to decrease the risk of antibiotic-resistant bacteria, has resulted in an increased interest in plant additives [3]. The bioactive compounds of wild plants have shown great promise for enhancing animal production due to their numerous biological activities that provide health benefits to animals [4,5]. 

Polyphenols, which are a wide range of secondary metabolites synthesized by plants, have antioxidant, anti-inflammatory, antimicrobial, and immunostimulant activities [6,7], whose types and quantities, like essential oils, are linked to the environmental and ecological characteristics in which the plants grow [8]. The bioavailability of polyphenols, measured as molecules absorbed in the small intestine, varies from 5% to 10% as observed by Faria et al. [9], due to their chemical structure such as esters, glycosides or polymers, and their association with cell wall constituents such as arabinoxylans, proteins, organic acids and lipids [10]. Dietary polyphenols are modified during absorption via hydrophilic conjugation (i.e., they are methylated, glucuronidated and sulphated), in the small intestine and later in the liver. Conjugation mechanisms are highly efficient, and aglycones are generally either absent or present in low concentrations in blood [6].

In food-producing animals, there is a clear relationship between the onset of several inflammatory and infectious diseases and the reduction of the antioxidant status [11]. Oxidative stress is also implicated in several pathological conditions that negatively affect animal health, welfare, and productive parameters [12]. At weaning, piglets are particularly susceptible to diseases and to oxidative stress related to several environmental and physiological factors. Antioxidant status declines after weaning, increasing piglet morbidity and mortality [9]. A recent study reported that a piglet’s antioxidant status after weaning is a reliable marker of their health status [13]. Previous studies have reported that dietary plant polyphenols in piglets enhance blood antioxidant status [14,15]. However, some studies report no improvements in the markers related to antioxidant status due to supplementation with polyphenols [16,17].

Considering that polyphenols can be conjugated, the determination of the antioxidant reserve could be an important tool to better understand the real antioxidant status of pigs and help to identify plants with high in vivo antioxidant activity. In fact, determination of the antioxidant reserve can be used to identify the amount of antioxidants stored in the organism, which could be released to counteract oxidative stress. However, further investigations would be necessary, using the Kit Radicaux Libres test (KRL™ Kirial International, Couternon, France), which evaluates the antioxidant status of an organism by testing the antioxidant defense systems and has been validated in pigs, showing a high sensitivity in detecting the total antioxidant activity [18,19,20,21].

The aim of the present study was to assess the total antioxidant activity and antioxidant reserves using the KRL test in weaned piglets fed various natural antioxidants. The influence of dietary antioxidants from plant extracts on the well-being of weaned piglets and their in vivo effects was also investigated.

## 2. Materials and Methods

### 2.1. Animals and Experimental Design

Procedures involving animals were carried out in accordance with the European Communities Council Directive (86/609/EEC, 1986) and approved by the Italian Ministry of Health (Law No. 116/92).

The study was conducted on a commercial farrow-to-wean farm in North Eastern Italy. For the experimental trial, one hundred and sixty piglets (Topics × Tempo) weaned at 28 days of age were randomly selected. Piglets were housed in four post-weaning environmentally controlled rooms containing eight pens, with five piglets per pen. The experiment was a completely randomized block design. Piglets were allocated randomly on the basis of litter, sex and live weight to one of four dietary treatments. The control group (C) received a commercial diet (Ferrero Mangimi S.P.A., Farigliano, Cuneo, Italy), and the experimental groups received the same diet supplemented with *Boswellia serrata* Roxb. extract (BOS); *Uncaria tomentosa* Willd. DC. and *Tanacetum parthenium* (L.) Sch. Bip. extracts (UT); or plant extract containing polyphenols (AOX).

The *Boswellia serrata* Roxb. dry hydroalcoholic extract (A.C.E.F., Fiorenzuola d’Arda, Piacenza, Italy) was added to the diet at a dosage of 500 mg/kg feed, and the extract was titrated at 65% of boswellic acid and at 2% of 3-acetyl-11-keto-β-boswellic acid (AKBA). The *Uncaria tomentosa* Willd. DC. and *Tanacetum parthenium* (L.) Sch. Bip. dry hydroalcoholic extracts (A.C.E.F., Fiorenzuola d’Arda, Piacenza, Italy) were added to the diet at dosages of 200 and 50 mg/kg feed, respectively. The *U. tomentosa* extract was titrated at 3% in total oxindole alkaloid, and the *T. parthenium* extract was titrated at 0.5% in parthenolide. The determination of the active compounds of *U. tomentosa*, *T. parthenium* and *B. serrata* was performed by the supplier (A.C.E.F. Fiorenzuola d’Arda, Italy) using high-performance liquid chromatography (HPLC) [12,13,14,15,16,17,18,19,20,21,22,23,24].

The polyphenol plant extract, composed of *Asteraceae, Verbenaceae* and *Limiaceae* was added to the diet at a dosage of 225 mg/kg feed. The AOX extract (Activ’XXS Inside^®^ supplied by Lara-Spiral, Couternon, France) was analyzed by HPLC and contained 25% polyphenols, including quercetin (3.4%), glycosylated quercetin (5.1%), chlorogenic acid (0.5%), chicoric acid (5.1%) and phenylpropanoids (5%) [25]. The supplementation rates of the selected plant extracts were based on the producer’s indications regarding the dosage to be employed for humans to obtain health effects.

The composition and nutrient content of the basal diets were formulated to meet the National Research Council’s nutrient requirements for piglets [26] and is reported in Table 1. The piglets were fed diets and water for ad libitum consumption.

A vitamin–trace mineral premix provided the following nutrients per kilogram of diet: vitamin A 15,000 UI; vitamin D3 2000 UI; vitamin E 250 mg; vitamin K3 2 mg; choline chloride 390 mg; folic acid 3 mg; niacin 50 mg; calcium d-pantothenate 10 mg; vitamin B1 10 mg; vitamin B6 10 mg; vitamin B12 0.050 mg; biotin 0.20 mg; Fe 384 mg; Cu 162 mg; Mn 80 mg; Zn 100 mg; Se 0.30 mg; I 1.5 mg; benzoic acid 5000 mg.

### 2.2. Data Collection

The experimental trial lasted 28 days. The health status and clinical signs of the animals were recorded daily. Piglets were weighed at 0, 18 and 28 days of the experimental trial. Amounts of feed offered and refused were recorded daily in order to estimate the feed intake of the piglets. These data were used to calculate the average daily gain (ADG) and feed conversion ratio (G:F) of each pen. 

### 2.3. Sampling

From 16 randomly selected piglets per treatment (2 piglets/pen), fasting blood samples were obtained at the beginning of the trial, and at 18 and 28 days, by anterior vena cava puncture. The blood samples were collected in 10 mL vacutainer glass tubes containing EDTA (Venoject^®^, Terumo Europe N.V., Leuven, Belgium) and immediately placed on ice. The samples were then stored at 4 °C until the analyses, which were performed within 24 h of collection.

### 2.4. Antioxidant Capacity of Plant Extract

The effects of the plant extracts on the sensitivity to free radical aggression were tested with KRL™, which measures the capacity of the control blood to withstand free-radical-induced hemolysis [21]. was used as the biological medium (Euro-Bio Snc, Brescia, Italy) and Trolox^®^ as a reference. Diluted blood samples with and without different doses of the final products were submitted to radical attack using the same method as in the subsections above. Preliminary cytotoxic assays without the addition of 2.2’-azobis (2-amidinopropane) dihydrochloride (AAPH; Kirial International, Dijon, France) were also conducted. Hemolysis was recorded using a 96-well microplate reader, and the results were expressed in KRL values. The kinetics of the control blood resistance to hemolysis were determined by continuous monitoring of changes in absorbance at 450 nm. Half hemolysis time (HT_50_) was retained for group comparisons. 

The results were expressed in mg Trolox equivalents per 100 g of product. A range from 0 to 1000 μmol/L of Trolox^®^ (MW 250.29 g/mol, Sigma-Aldrich, Saint Louis, MO, USA) enabled us to standardize the global antioxidant capacity of the product compared to vitamin E. The natural extracts were tested in triplicate in independent analyses.

### 2.5. Blood Analyses

#### 2.5.1. Total Antioxidant Activity

The KRL™ test evaluates the antioxidant status of an organism by testing the antioxidant defense systems [20,21]. The antiradical potential of both the whole piglet blood and the red blood cells (RBCs) was evaluated using a KRL™ test, which tests blood resistance based on free-radical-induced hemolysis. Whole blood and RBC samples, under orbital shaking, were each submitted to an isotonic saline solution to organic free radical aggression (AAPH) produced at 37 °C. The extracellular and intracellular antioxidant defenses contribute to maintaining blood cell membrane integrity and function, until cell lysis. Hemolysis was recorded using a 96-well microplate reader (KRL Reader™—Kirial International, Couternon, France) by measuring the optical density decay at 450 nm. Results were expressed as the time required to reach 50% hemolysis. Half-hemolysis times (HT_50_) for whole blood and red blood cells (RBCs) were expressed in minutes, referring to the whole blood and RBC resistance to free radical attack. The measurement of HT_50_ was taken to be representative of the overall defenses against free radicals (KRL value). Intra- and inter-assay coefficients of variation of the KRL test were 2.5% and 4%, respectively.

#### 2.5.2. Antioxidant Reserves

The release of antioxidant reserves in blood was evaluated by the KRL test, using the RESEDA™ (RESErves Défenses Antioxydantes) test (Spiral Laboratories, 21560 Couternon, France) [27] on fresh ethylenediamine tetraacetic acid blood from the piglets, sampled at 28 days. To evaluate the effects of the dietary natural extract supplementation on the release of antioxidant reserves, several enzymes were added: reserve R1 (hydrolysis with glucosidase), R2 (hydrolysis by means of sulfatase) and R3 (hydrolysis with glucuronidase). Whole blood samples were then submitted to an isotonic saline solution to organic free radicals (AAPH) produced at 37 °C, and hemolysis was recorded using a 96-well microplate reader by measuring the optical density decay at 450 nm. Results were expressed as the percentage increase of KRL value after hydrolysis with respect to the KRL value of whole blood.

### 2.6. Statistical Analysis

The data on growth performance were analyzed by one-way analysis of variance (ANOVA) to show the effect of treatment, with the pen as the experimental unit. The data on the KRL values of whole blood and RBCs were assessed by two-way repeated measures ANOVA to show the effects of treatment, time, and their interactions. The KRL values of whole blood and RBCs at the beginning of the trial were used as covariates. The data on antioxidant reserves were analyzed by ANOVA to show the effect of treatment. The piglet was considered as the experimental unit for blood analyses. Data were presented as means ± SEM, and a value of *p* < 0.05 was used to indicate statistical significance.

## 3. Results

### 3.1. In Vitro Antioxidant Activity of Natural Extracts

Figure 1 reports the in vitro antioxidant activity of natural extracts. Our results highlight that all the extracts showed an in vitro antioxidant capacity, which increased linearly with the dose of the extract, until a concentration of 0.1 g/L. Low antioxidant activity was observed for the *B. serrata* extract and for the *U. tomentosa* and *T. parthenium* mixture (15.43 vs. 46.72 eq. Trolox/100 g of product, respectively). The polyphenol plant extract mixture exhibited a high antioxidant activity with a value of 1092.66 eq. Trolox/100 g of product. 

### 3.2. Productive Parameters

The data on piglets’ growth performance in relation to dietary supplementation with the selected natural extracts are reported in Table 2. The piglets were healthy throughout the experimental period. A mortality rate of 5% was observed in the BOS group, while mortality was not recorded in other experimental groups. No differences (*p* > 0.05) in final weight and ADG were observed among the experimental groups. The feed conversion ratio was also unaffected (*p* > 0.05) by dietary treatments with natural extracts containing polyphenols.

### 3.3. Total Antioxidant Activity

The KRL values of the whole blood and red blood cells of piglets in relation to dietary treatment and sampling time are reported in Figure 2. No significant differences (*p* > 0.05) of whole blood KRL values were observed in relation to the sampling time and dietary treatment. A significant interaction (*p* < 0.05) between sampling time and dietary treatments was observed for whole blood. A significant effect (*p* < 0.01) of dietary treatments was observed on the KRL value of the red blood cells. Sampling time did not affect (*p* > 0.05) this parameter. A significant interaction (*p* < 0.05) between sampling time and dietary treatments was also observed.

The total antiradical activities of the whole blood samples showed the total antioxidant defenses of the organism at the time of sampling. The KRL analysis performed on RBCs refers to the intracellular defense status and, considering that the average RBC life span is about 60–85 days, these data reflected the antioxidant defense of the last one to two months.

The delta KRL values of whole blood showed a significantly greater increase of the total antioxidant status of the piglets from the BOS and AOX groups than in the UT and C groups (+30.7 BOS; +27.7 AOX vs. +17.81 UT +13.30 C; *p* = 0.002) between the two sampling times.

The delta KRL values of RBCs showed a significant increase in the total antioxidant status of piglets from AOX groups compared to the UT and BOS groups (+22.2 AOX vs. +9.90 UT +9.4 BOS; *p* = 0.016) between the two sampling times. These data reveal that the systemic antioxidant status of the piglets was improved by the AOX and *B. serrata* extract dietary supplementation.

### 3.4. Antioxidant Reserves

Figure 3 shows the piglets’ blood antioxidant reserves in relation to dietary treatments at the end of the trial. The RESEDA antioxidant reserves are expressed as the percentage KRL value of whole blood released by hydrolysis with glucosidase, sulfatase and glucuronidase, in respect to the KRL value of whole blood.

The present data show that piglets fed diets supplemented with *U. tomentosa* and *T. parthenium* extracts had higher (*p* < 0.05) blood reserves as glucosides than the C and BOS groups (Figure 3A). The blood reserves as glucosides of the AOX group did not differ (*p* > 0.05) from the other experimental groups.

The blood antioxidant reserves as sulphates were higher (*p* < 0.05) in piglets fed *U. tomentosa* and *T. parthenium* extracts than the controls. The blood reserves as sulphates in the BOS and AOX groups did not differ (*p* > 0.05) from the other experimental groups (Figure 3B).

The blood antioxidant reserves as glucuronides were higher (*p* < 0.05) in piglets fed the *U. tomentosa* and *T. parthenium* extracts than in the *B. serrata* group. The blood reserves as glucuronides in the C and AOX groups did not differ (*p* > 0.05) from the other experimental groups (Figure 3C).

## 4. Discussion

In the UT extract mixture, the antioxidant activity was related to the polyphenol content of *U. tomentosa* [28] and the polyphenol and parthenolide content of *T. parthenium,* which showed a free radical scavenging activity. The antioxidant activity of the *B. serrata* extract may be related to its acetyl-11-keto-beta-boswellic acid (AKBA) content, which is one of the active principles of boswellic acids that has shown antioxidant activity. In fact, as has been observed in cultured fibroblasts, AKBA is able to decrease oxidative stress, reducing intracellular reactive oxygen species (ROS) concentrations and increasing both superoxide dismutase (SOD) and glutathione peroxidase (GPx) activity in rat serum [29]. The mechanism of action of AKBA is related to its superoxide scavenging properties through the Nrf2/HO-1 pathway [30].

As observed in our previous study, the KRL assay showed a high sensitivity in detecting the antioxidant activity of the natural extract, and could be a promising approach to screening plant extracts in order to determine the global antioxidant activity in a biological system [31].

Recent studies have reported that weaning can induce oxidative stress in pigs, thus making piglets more susceptible to diseases and growth depression [32,33]. Body antioxidants delay or inhibit oxidation processes [34], and several studies have reported the positive effects of dietary plant extracts containing polyphenols on the antioxidant status in piglets [35,36].

The present data show that dietary supplementation with all the selected plant extracts did not affect the growth performance of post-weaned piglets. The effects of plant extract dietary supplementation on growth parameters in pigs are contradictory. The present data agree with a previous study by Fiesel et al. [37], who reported no effects on piglets’ growth performance due to dietary supplementation with 1% grape seed and grape marc meal extract or spent hops, which are rich in polyphenols. An improvement in gain-to-feed ratio was observed.

Xu et al. [38] showed no growth performance improvement in piglets fed 400 mg/kg or 800 mg/kg of apple polyphenol feed. In contrast, Corino et al. [39] showed that a dietary biotechnological extract from *Ajuga reptans* L. containing 5 or 10 mg/kg feed of teupolioside improved the ADG and final weight of weaned piglets. Devi et al. [40] also reported that dietary supplementation with a 0.05% clove, cinnamon and fenugreek mixture increased piglets’ ADG compared to piglets fed antimicrobials and an organic acid supplement.

In our study, dietary supplementation with the selected plant extracts did not affect the total antioxidant activity of whole blood. The detected KRL values were in line with the reference values of post-weaned piglets [41]. Significant differences in the red blood cells in relation to dietary treatment were observed. A greater delta KRL value of RBCs in the AOX group was observed, in comparison to the UT and BOS groups. This effect could be due to the high antioxidant activity of the mix of plant antioxidants in AOX, as shown by the KRL test.

In a previous work [19], we reported that long-term dietary supplementation from weaning to slaughter with *Verbenaceae* extract, containing polyphenols, tended to improve the whole blood KRL value of pigs. It is possible that this difference is related to the shorter dietary supplementation, or to the different dosage and composition of the plant extract employed.

To the best of our knowledge, no previous study has reported the effects of dietary supplementation with plant extracts on blood antioxidant reserves in weaned piglets. The present data show that the piglets in the *U. tomentosa* and *T. parthenium* mixture group had higher blood reserves as glucuronides, sulphates, and glucosides, which may be mobilized by the piglets to combat oxidative stress. The lowest values reported were for the *B. serrata* extract group, which also showed the lowest antioxidant activity in the KRL test. However, the KRL analysis showed a high antioxidant activity for the AOX extract.

Discrepancies between in vitro and in vivo data are common. Plant polyphenols appear mainly as O-glycosides and are hydrolyzed into aglycones in the acidic environment of the stomach. After absorption, they support the systemic antioxidant or are enzymatically converted into O-glucuronides and O-sulfates in the liver. In addition, gut bacteria have a glucuronidase capacity, and reconvert polyphenol into available molecules, thus enhancing polyphenol bioavailability [42].

All the data on the piglets’ antioxidant status suggest that AOX and BOS supplements sustain the systemic antioxidant status, as shown by the whole blood delta KRL value from the two sampling times, and very little is converted into antioxidant reserves. It seems that the UT supplement is mainly converted into blood reserves in the form of glucuronides, sulphates and glucosides, without affecting the piglet’s blood KRL value. The present data show that several polyphenol molecules from the selected plant extract play different roles in the organism that are related to the complexity of the phenolic metabolism and the association with the intestinal microbiome, which is not yet fully understood.

## 5. Conclusions

The present work provides very interesting results regarding the effects of the dietary supplementation of several natural extracts on the antioxidant status of weaned piglets. The biological test KRL proved to be an extremely sensitive tool to evaluate the antioxidant capacity of these extracts. This is the first study that analyzes the effects of the dietary supplementation of natural extracts on the important antioxidant reserves of blood in piglets. Further studies are needed to better understand the role that different reserves play in conditions of acute and chronic oxidative stress, and to deepen the knowledge about the effects of other wild plant extracts containing polyphenols on this important parameter.

## Figures and Tables

**Figure 1 antioxidants-10-00702-f001:**
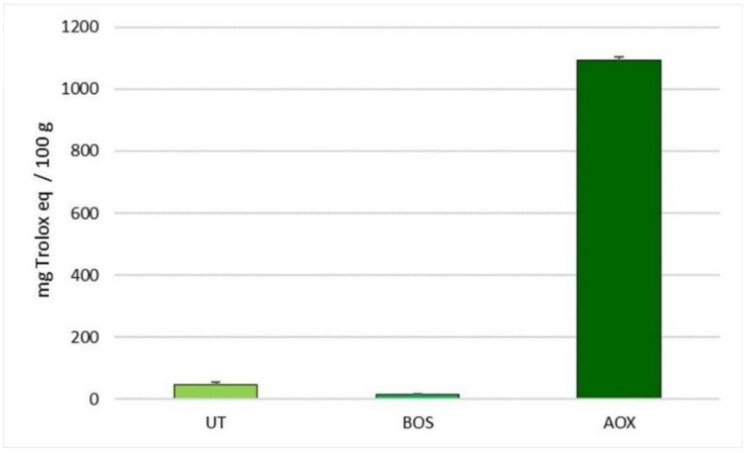
In Vitro antioxidant activity of the selected natural extracts measured with biological test KRL (concentration 0.1 g/L). Data are reported as mean ± SEM; *n* = 3. BOS, *B. serrata* extract; UT, *U. tomentosa* and *T. parthenium* extracts; AOX, antioxidant plant extract mixture.

**Figure 2 antioxidants-10-00702-f002:**
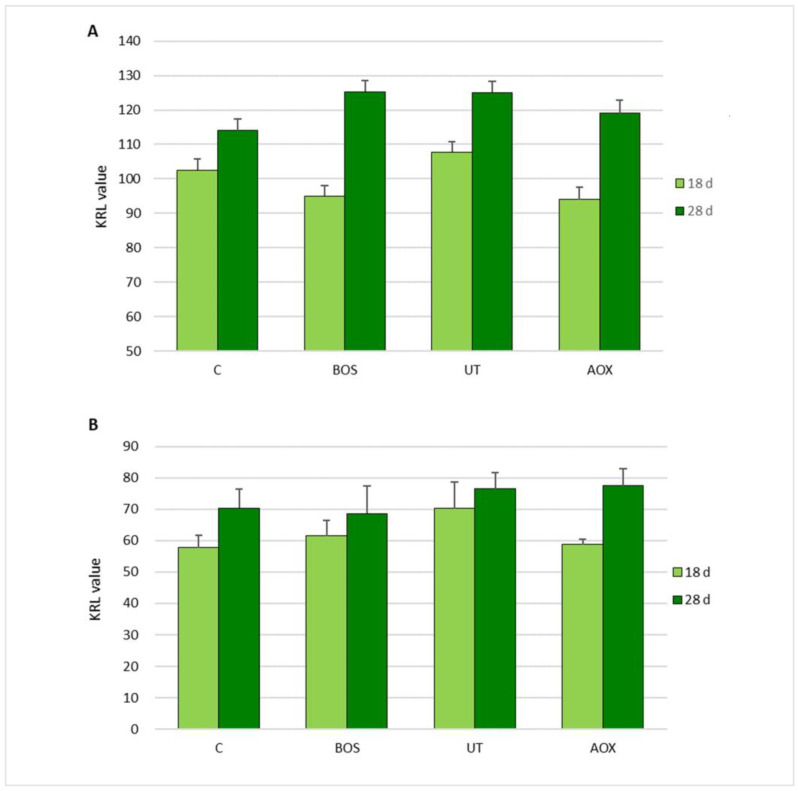
The KRL value of piglets’ whole blood (**A**) and red blood cells (**B**), covaried by KRL values at the beginning of the trial, in relation to dietary treatments and sampling time. Data are reported as mean ± SEM; *n* = 16. C, control group; BOS, diet supplemented with *B. serrata* extract; UT, diet supplemented with *U. tomentosa* and *T. parthenium* extracts; AOX, diet supplemented with antioxidant plant extract mixture. For whole blood: Treatment *p* = 0.123; Time *p* = 0.543; Time × Treatment interaction, *p* = 0.002. For red blood cells: Treatment *p* = 0.003; Time *p* = 0.795; Time × Treatment interaction *p* = 0.014.

**Figure 3 antioxidants-10-00702-f003:**
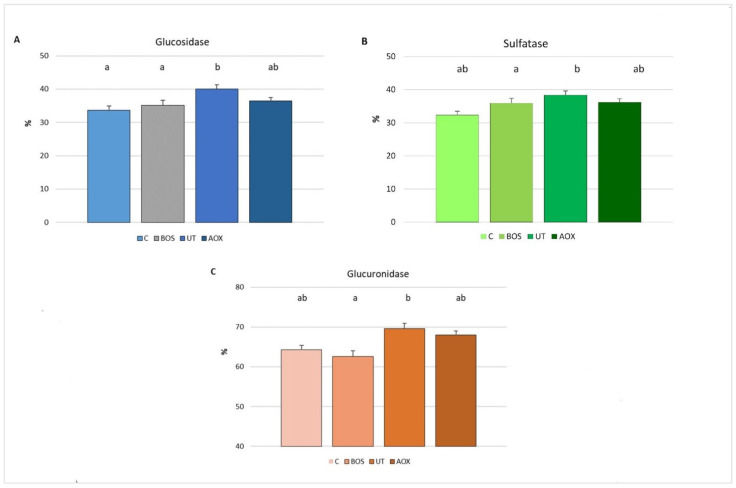
Piglets’ blood antioxidant reserves in relation to dietary treatments, expressed as percentage of KRL value of whole blood released by enzymes ((**A**) hydrolysis with glucosidase; (**B**) hydrolysis with sulfatase and (**C**) hydrolysis with glucuronidase). Data are reported as mean ± SEM; *n* = 16. C, control group; BOS, diet supplemented with *B. serrata* extract; UT, diet supplemented with *U. tomentosa* and *T. parthenium* extracts; AOX, diet supplemented with antioxidant plant extract mixture. ^a,b^ Means with different superscripts differ (*p* < 0.05).

**Table 1 antioxidants-10-00702-t001:** Nutrient content of the commercial diet ^1^.

Nutrients	%, As Fed
Crude protein	16.00
Fat	5.5
Crude fiber	3.5
Ash	5.5
Calcium	0.62
Phosphorus	0.66
Lysine	1.24
Methionine	0.48

^1^ Composition: barley, extruded wheat, barley flakes, whey powder, heat-treated soybeans, corn, fishmeal, potato protein, dicalcium phosphate, lignocellulose, coconut oil, soybean oil, animal fats, sodium chloride.

**Table 2 antioxidants-10-00702-t002:** Growth performances of piglets fed the control diet (C) and the diet supplemented with *B. serrata* extract (BOS), *U. tomentosa* and *T. parthenium* extract mixture (UT), and the plant extract mixture (AOX).

Growth Performances	Dietary Treatments				*p*
	C	BOS	UT	AOX	
Body weight, kg					
—at weaning	9.44 ± 0.19	9.52 ± 0.16	9.50 ± 0.18	9.37± 0.18	0.966
—at 28 d after weaning	21.53 ± 0.48	20.40 ± 0.47	21.25 ± 0.45	20.70 ± 0.45	0.307
ADG, g/d	431 ± 14.5	388 ± 15.0	419 ± 13.9	404 ± 15.1	0.188
FCR, kg/kg	1.56	1.64	1.48	1.59	0.499

Data are reported as mean ± SEM; *n* = 8. C, control group; BOS, diet supplemented with *B. serrata* extract; UT, diet supplemented with *U. tomentosa* and *T. parthenium* extracts; AOX, diet supplemented with antioxidant plant extract mixture; ADG, average daily gain; FCR, feed conversion ratio.

## Data Availability

The data presented in this study are available on request from the corresponding author.

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
