# Peer review of "Dietary Plant Extracts Improve the Antioxidant Reserves in Weaned Piglets"

_antioxidants, 2021, doi:10.3390/antiox10050702_

Round 1

Reviewer 1 Report

Studies assessing the impact of polyphenol-rich dietary supplements on the antioxidant status of newly weaned piglets, and thus on the health, welfare and productivity of growing animals, are of utmost relevance. The specific study evaluates the interaction between the three crude polyphenolic commercial extracts and the period after weaning (18 and 28 days), and compares the antiradical potential of whole blood and RBC by using the biological test based on free radical induced haemolysis. The special importance of this paper stems from the fact that authors take into account the antioxidant reserves of whole blood (as glucuronides, sulphates and glucosides), which may be a quantitative marker to differentiate the antioxidant activity of dietary plant extracts for young pigs.

The manuscript is well written with clear objectives. Background literature is well reviewed while the gap in knowledge being filled by the present work is highlighted. Methodology employed is fully described. Results are presented in a comprehensible way. In the discussion section all main findings are interpreted and related to those of previous studies. The need for further studies on the relevance of different reserves in conditions of oxidative stress is also convincingly justified.

Author Response

Response to Referee: 1

Studies assessing the impact of polyphenol-rich dietary supplements on the antioxidant status of newly weaned piglets, and thus on the health, welfare and productivity of growing animals, are of utmost relevance. The specific study evaluates the interaction between the three crude polyphenolic commercial extracts and the period after weaning (18 and 28 days), and compares the antiradical potential of whole blood and RBC by using the biological test based on free radical induced haemolysis. The special importance of this paper stems from the fact that authors take into account the antioxidant reserves of whole blood (as glucuronides, sulphates and glucosides), which may be a quantitative marker to differentiate the antioxidant activity of dietary plant extracts for young pigs.The manuscript is well written with clear objectives. Background literature is well reviewed while the gap in knowledge being filled by the present work is highlighted. Methodology employed is fully described. Results are presented in a comprehensible way. In the discussion section all main findings are interpreted and related to those of previous studies. The need for further studies on the relevance of different reserves in conditions of oxidative stress is also convincingly justified.

The authors would like to thank the referee for the positive comments on our manuscript.

Reviewer 2 Report

The authors propose a manuscript titled “Dietary plant extracts improve the antioxidant reserves in weaned piglets”. The article is well structured. In particular, this study takes into consideration a topic aspect about the reducing use of antibiotics in livestock, to contain antibiotic resistance, in favor of natural substances that arrive from the biological activities of plant extracts and their antioxidant activity. The study assesses the total antioxidant activity and antioxidant reserves using the KRL test on one hundred and sixty piglets. I appreciate the original idea of the work which with a few revisions will convince me and the editor to publish it on Journal.

Introduction

- Please give a brief explanation of what hydroponic systems means in order to introduce the reader to the subject of the manuscript. Some people is aware of this interesting sector.

- Choose two references for these statements

- Lines 33-34. “Natural extracts are used in animal nutrition as sustainable additives to improve growth performance and health.

- Lines 37-39. Please add the term wild in this way and add a reference: “The bioactive compounds of wild plants have shown great promise for enhancing animal production due to their numerous biological activities that provide health benefits to animals [2, Perrino et al. 2021, Maina et al. 2020]”.

References to be added:

  • Maina, S.; Misinzo, G.; Bakari, G.; Kim, H.-Y. Human, Animal and Plant Health Benefits of Glucosinolates and Strategies for Enhanced Bioactivity: A Systematic Review. Molecules2020, 25, 3682.
  • Perrino, E. V.; Musarella, C. M.; Magazzini, P. Management of grazing Italian river buffalo to preserve habitats defined by Directive 92/43/EEC in a protected wetland area on the Mediterranean coast: Palude Frattarolo, Apulia, Italy. Euro-Mediterr. J. Environ. Integr. 2021, 6, 32

- Lines 40-41. Please complete the period with a crucial aspect, in the follow way: “Polyphenols, which are a wide range of secondary metabolites synthesized by plants, have antioxidant, anti-inflammatory antimicrobial and immunostimulant activities [3,4], whose types and quantities, like essential oils, are linked of the environmental and ecological characteristics in which the plants grow [Perrino et al. 2021]

  • Perrino, E.V.; Valerio, F.; Gannouchi, A.; Trani, A.; Mezzapesa, G. Ecological and Plant Community Implication on Essential Oils Composition in Useful Wild Officinal Species: A Pilot Case Study in Apulia (Italy). Plants 2021, 10, 574. https://doi.org/10.3390/plants10030574
  1. Materials and Methods

Well done. Only few suggestions:

  • Line 75. In italic “…1 Animals”
  • Lines 88 and 91. Please when cited for the first time the scientific name of the plant species use the complete way considering also the author name. Remember that the name of genus and species goes in italic unlike the author. E.g. Uncaria tomentosa DC. …not Dc.
  • Line 88. Boswellia serrata ….? Complete
  • Line 91. Tanacetum parthenium ….? Complete
  • Lines 98-99. Lamiaceae is the current name of Labiatae (Lamiaceae=Labiatae). So cut the word Labiatae.
  1. Results

Well done, the figures are clear

Lines 264, 265, 269, 272. In the internationa audiance the plant species must be reported in the complete way, not only genus (Uncaria, Tanacetum, Boswellia). I suggest: U. tomentosa, T. parthenium, B. serrata

  1. Discussion

Lines 277-278. Follow the previous comment.

Figure 3. I would chosen different colors than variants of green colours to better understand the differences between different types of hydrolysis.

Line 312. Ajuga reptans. See my previous comment, …for the first time the complete name of the plant species.

Line 330. U. tomentosa, T. parthenium

  1. Conclusion

I am full agree with this statement “Further studies are needed to better understand the role that different reserves play in conditions of acute and chronic oxidative stress.”, but I suggest to spend two words about that other wild plants could have contents in polyphenols or other chemical compounds useful for the object of the study.

Author Response

The authors would like to thank the referee for the constructive comments and suggestions on our manuscript. Those comments are very helpful for revising and improving our paper. The suggested changes have been made and are reported in the reviewed article in a red font.

Introduction

Please give a brief explanation of what hydroponic systems means in order to introduce the reader to the subject of the manuscript. Some people is aware of this interesting sector.

No explanation on hydroponic systems has been inserted because it is poor related to the paper topics.

  1. Choose two references for these statements Lines 33-34. “Natural extracts are used in animal nutrition as sustainable additives to improve growth performance and health.

New lines 33-34: As suggested, the references have been added in the text

  1. Lines 37-39. Please add the term wild in this way and add a reference: “The bioactive compounds of wild plants have shown great promise for enhancing animal production due to their numerous biological activities that provide health benefits to animals[2, Perrino et al. 2021, Maina et al. 2020]”. References to be added:
  2. Maina, S.; Misinzo, G.; Bakari, G.; Kim, H.-Y. Human, Animal and Plant Health Benefits of Glucosinolates and Strategies for Enhanced Bioactivity: A Systematic Review. Molecules2020, 25, 3682.
  3. Perrino, E. V.; Musarella, C. M.; Magazzini, P. Management of grazing Italian river buffalo to preserve habitats defined by Directive 92/43/EEC in a protected wetland area on the Mediterranean coast: Palude Frattarolo, Apulia, Italy. Euro-Mediterr. J. Environ. Integr. 2021, 6, 32

New lines 39: The firs reference has been added in the text as requested, the second one has not been inserted because one seems to be poorly related to the sentence’s topics.

Lines 40-41. Please complete the period with a crucial aspect, in the follow way: “Polyphenols, which are a wide range of secondary metabolites synthesized by plants, have antioxidant, anti-inflammatory antimicrobial and immunostimulant activities [3,4], whose types and quantities, like essential oils, are linked of the environmental and ecological characteristics in which the plants grow [Perrino et al. 2021]

New Line 41-43: the suggested change has been made.

Materials and Methods

Line 75. In italic “…1 Animals”

New Line 77 : the suggested change has been made.

Lines 88 and 91. Please when cited for the first time the scientific name of the plant species use the complete way considering also the author name. Remember that the name of genus and species goes in italic unlike the author. E.g. Uncaria tomentosa DC. …not Dc.

New Line 77 : the suggested change has been made.

Line 88. Boswellia serrata ….? Complete

New line 88: The suggested change has been made

Line 91. Tanacetum parthenium ….? Complete

New line 88: The suggested change has been made

Lines 98-99. Lamiaceae is the current name of Labiatae (Lamiaceae=Labiatae). So cut the word Labiatae.

New line Line 101: The suggested change has been made

Results

Lines 264, 265, 269, 272. In the internationa audiance the plant species must be reported in the complete way, not only genus (Uncaria, Tanacetum, Boswellia). I suggest: U. tomentosaT. partheniumB. serrata

The suggested change has been made throughout the manuscript

Discussion

Lines 277-278. Follow the previous comment.

The suggested change has been made throughout the manuscript

Figure 3. I would chosen different colors than variants of green colours to better understand the differences between different types of hydrolysis.

Figure 3. The suggested change has been made.

Line 312. Ajuga reptans. See my previous comment, …for the first time the complete name of the plant species.

New line 317 The suggested change has been made.

Line 330. U. tomentosaT. parthenium

New line 336: The suggested change has been made.

Conclusion

I am full agree with this statement “Further studies are needed to better understand the role that different reserves play in conditions of acute and chronic oxidative stress.”, but I suggest to spend two words about that other wild plants could have contents in polyphenols or other chemical compounds useful for the object of the study.

The sentence has been revised in: “Further studies are needed to better understand the role that different reserves play in conditions of acute and chronic oxidative stress and to deep the knowledge about the effect of other wild plant extracts, containing polyphenols, on this important parameter.”